# Genetically Engineered Mouse Models of Liver Tumorigenesis Reveal a Wide Histological Spectrum of Neoplastic and Non-Neoplastic Liver Lesions

**DOI:** 10.3390/cancers12082265

**Published:** 2020-08-13

**Authors:** Katja Steiger, Nina Gross, Sebastian A. Widholz, Roland Rad, Wilko Weichert, Carolin Mogler

**Affiliations:** 1Institute of Pathology, School of Medicine, Technical University of Munich (TUM), 81675 Munich, Germany; wilko.weichert@tum.de (W.W.); carolin.mogler@tum.de (C.M.); 2Comparative Experimental Pathology, School of Medicine, Technical University of Munich (TUM), 81675 Munich, Germany; 3Member of the German Cancer Consortium (DKTK), Partner Site Munich, 81675 Munich, Germany; roland.rad@tum.de; 4Center for Translational Cancer Research (Transla TUM), School of Medicine, Technical University of Munich (TUM), 81675 Munich, Germany; nina.gross@tum.de (N.G.); sebastian.widholz@tum.de (S.A.W.); 5Institute of Molecular Oncology and Functional Genomics, School of Medicine, Technical University of Munich (TUM), 81675 Munich, Germany; 6Department of Internal Medicine II, Klinikum rechts der Isar, School of Medicine, Technical University of Munich (TUM), 81675 Munich, Germany

**Keywords:** GEMM, liver tumorigenesis, liver lesions, histology

## Abstract

Genetically engineered mouse models (GEMM) are an elegant tool to study liver carcinogenesis in vivo. Newly designed mouse models need detailed (histopathological) phenotyping when described for the first time to avoid misinterpretation and misconclusions. Many chemically induced models for hepatocarcinogenesis comprise a huge variety of histologically benign and malignant neoplastic, as well as non-neoplastic, lesions. Such comprehensive categorization data for GEMM are still missing. In this study, 874 microscopically categorized liver lesions from 369 macroscopically detected liver “tumors” from five different GEMM for liver tumorigenesis were included. The histologic spectrum of diagnosis included a wide range of both benign and malignant neoplastic (approx. 82%) and non-neoplastic (approx. 18%) lesions including hyperplasia, reactive bile duct changes or oval cell proliferations with huge variations among the various models and genetic backgrounds. Our study therefore critically demonstrates that models of liver tumorigenesis can harbor a huge variety of histopathologically distinct diagnosis and, depending on the genotype, notable variations are expectable. These findings are extremely important to warrant the correct application of GEMM in liver cancer research and clearly emphasize the role of basic histopathology as still being a crucial tool in modern biomedical research.

## 1. Introduction

Genetically engineered mouse models (GEMM) are an elegant tool to study carcinogenesis in vivo. In the last few years, GEMM have become one of the most relevant mouse models in biomedical research [1,2,3,4,5,6,7,8]. Usually GEMM are used to study the morphomolecular relationships of human diseases after modelling genetic alterations associated with the human counterpart [9]. When newly designed GEMM are published for the first time, a detailed histopathological phenotyping is paramount to correctly classify macroscopic findings. Unfortunately, proper phenotyping is not always provided, or the assessment of the respective phenotype is not always correct, due to a lack of expertise or access to a histopathological (core) unit [10]. This is particularly problematic for the scientific community, because the use of generally accepted pathology nomenclature for unexpected and novel findings leads to publications that can be interpreted by readers, including other pathologists [11]. Additionally, this is important, since histopathological phenotypes in GEMM can be very tricky to diagnose, specifically in the liver “tumor” setting, mirroring the situation in human liver pathology [12,13,14,15,16,17,18]. Additionally, although a histopathological classification of animal proliferative (and non-proliferative) liver lesions exists [19], these definitions, provided by the Society of (Veterinarian) Toxicopathology, are unknown to broad parts of the scientific community. Both circumstances might cause or support a weak rate of reproducibility of histopathological results published, even very frequently, in high ranking journals [11,20]. Chemically induced rodent models for liver tumorigenesis (especially the so-called Diethylnitrosamine DEN-models) are well characterized and present with a huge variety of proliferative liver lesions of both neoplastic and non-neoplastic origin [21,22,23]. The most widely accepted classification of proliferative and non-proliferative liver lesions originates from the INHAND consortium (International Harmonization of Nomenclature and Diagnostic Criteria for Lesions in Rats and Mice) [19]. Hepatic lesions that might be relevant for oncologic GEMM include alterations occurring as hepatocellular response (i.e., fatty change, hepatocellular hypertrophy, glycogen accumulation, karyocytomegaly, and multinucleated hepatocytes), inflammatory lesions (i.e., inflammatory cell infiltration of different types of inflammatory cells and fibrosis), non-neoplastic proliferative alterations (i.e., foci of cellular alteration, hepatocellular hyperplasia (regenerative/non-regenerative), Kupffer-cell and Ito cell hyperplasia, bile duct hyperplasia, cholangiofibrosis, oval cell hyperplasia) and neoplasms (i.e., hepatocellular adenoma and carcinoma, hepatoblastoma, cholangioma, cholangiocarcinoma, hepatocholangiocellular adenoma and carcinoma, benign Ito cell tumor, histiocytic sarcoma, hemangioma, and hemangiosarcoma).

The sequence of liver tumorigenesis has been investigated thoroughly and described morphologically in animal models of hepatic carcinogenesis [24,25]. In contrast, in GEMM, systematic comparative morphomolecular investigations are lacking.

The most frequently used GEMMs in hepatocellular carcinoma (HCC) research are based on liver specific overexpression of oncogenes or conditional knock-out of tumor-suppressor genes involved in hepatocarcinogenesis. Models using liver- specific overexpression of the transcription factor c-Myc are widely used in liver cancer research. While some authors carefully characterize the neoplastic lesions in these animals as “hepatocellular neoplasia with some features of HCC” [26], others describe HCC and “dysplastic nodules” [27] and again others consider all macroscopically detectable lesions as HCC [28]. In a different model with a well-known oncogene (KRAS) expressed under a liver-specific promotor (Albumin-Cre), one study describes HCC in all animals without mentioning potential precursor or other lesions in the livers of these animals [29], while a different study revealed the occurrence of dysplastic nodules and no HCCs [30]. The aim of this study was therefore to investigate if GEMM for liver tumorigenesis do harbor a comparably wide spectrum of liver lesions as in chemically induced mouse models and to apply a standardized classification scheme based on published criteria.

## 2. Materials and Methods

A total of 369 formalin-fixed paraffin embedded blocks (FFPE) from macroscopically detectable tumor nodules were obtained from five GEMM with genetic alterations in various oncogenic pathways. An AlbCre (Speer6-ps1^Tg(Alb-cre)21^) line was used to allow for conditional genetic manipulation. Animals were crossed in different combinations with animals with genetic modifications of the KRAS (Kras^tm4^), PTEN (Pten^flox^), TGFβR2 (TGFβR2^tm1^) and/or IDH1 (IDH1^tm2^)- allele. KRAS (*n* = 130), KRAS/PTEN (*n* = 26), PTEN (*n* = 82), PTEN/TGFβR2 (*n* = 111) and PTEN/IDH1 (*n* = 20) animals were analyzed, age and sex distribution for the different models are depicted in Appendix A. All samples were obtained from the Comparative Experimental Pathology (CEP) at the Institute of Pathology, Technical University Munich (TUM), one of the largest core facilities in Europe for comparative pathology. An adequate level of genetic modification/deletion of each mouse and model has been assured by the collaboration partners [31,32]. Animals were initially provided to our collaboration partners by the Welcome Trust Sanger Institute, Genome Campus, Hinxton, Cambridge, CB10 1SA, UK. Experiments were approved by the local ethical committees in both the UK and Germany (TV 55.2-2532.Vet_02-16-143, Reg. v. Oberbayern, year of approval included in number). Mice were all kept under standard laboratory conditions (12 h day/night cycle, water and standard diet ad libitum, no special diet). Only samples from animals originating from end-point studies were included. Samples from animals with unclear/insufficient extent of genetic knockdown were excluded from this study. For histopathological evaluation, H&E stainings were performed according to standard procedures and independently evaluated by two experienced comparative pathologists, one veterinary pathologist (K.S.) and one human surgical pathologist (C.M.), both with ample expertise in liver mouse pathology. Lesions were classified according to the published INHAND criteria [19]. If more than one lesion was microscopically detectable in one macroscopically described tumor, histological diagnoses were reported separately. If a clear diagnosis could not be made due to insufficient tissue quality, inadequate sampling (e.g., only tumor, no adjacent tissue) or if both pathologists were not able to agree on one diagnosis, lesions were classified as “unclassifiable”. The classification criteria for proliferative liver lesions were (according to Thoolen et al. [19] with slight modifications for clarification):

Focus of cellular alteration: no or only minimal compression of the surrounding liver tissue, sharply demarcated from the adjacent normal hepatocytes by the appearance and staining reaction of its cells, lobular architecture preserved.

Hepatocellular hyperplasia: comprised of slightly enlarged hepatocytes tinctorially seminal to surrounding parenchyma, minimal to mild compression of adjacent parenchyma, lobular architecture maintained

Bile duct hyperplasia: increased number of small, well-differentiated bile ducts arising in portal region.

Cholangiofibrosis: consisting of dilated to cystic bile ducts filled with mucus and debris, surrounded by inflammatory cell infiltrates and connective tissue, the epithelium was usually single layered, epithelial cells sometimes showed a certain degree of cellular pleomorphism.

Oval cell hyperplasia: single or double row of oval to round cells along sinusoids often forming small ductules streaming into the hepatic parenchyma

Hepatocellular adenoma: nodular lesion compressing (at least on two quadrants) the adjacent normal hepatocytes, sharply demarcated; loss of the normal lobular architecture with irregular growth pattern, liver plates often impinged obliquely on surrounding liver parenchyma.

Hepatocellular carcinoma: local infiltrating growth and/or lack of distinct demarcation, loss of normal lobular architecture, trabeculae with 3 or more cell-layers.

Bile duct adenoma: uniform and well-circumscribed neoplasm with a single layer of cuboidal cells, growing expansively with compression.

Cholangiocarcinoma: glandular structures lined by single or multilayered cuboidal or cylindrical cells, invading into vascular or lymphatic structures and/or surrounding parenchyma

Undifferentiated carcinoma: carcinoma of hepatocellular or cholangiocellular origin where a clear specification of the cell type of origin (based on H&E evaluation) was not possible.

Oval cell tumor: oval cell proliferation invading into the hepatic parenchyma with prominent cellular atypia and increased nuclear:cytoplasm ratio, accompanied by prominent spindle cells.

It has to be mentioned that we did not use the term “dysplastic nodule”, which is often applied to describe early changes in GEMM when they are compared to the human counterpart. These lesions have been termed “foci of cellular alterations” according to the available and accepted classification for murine lesions.

## 3. Results

A total of 369 slides were evaluated, containing one macroscopically detectable tumor each. Hereby, a total of 874 lesions (on average 2.37 lesions/slide) could be diagnosed histologically (range: 1–50 nodules/per slide). A complete list of histopathological diagnosis among the several models can be found in Table 1.

Among those 874 lesions, 714 lesions (approximately 81.7%) were classified as neoplastic. The diagnosis list included benign neoplastic lesions (hepatocellular or bile duct adenoma, *n* = 3 each; Figure 1A,B), potential premalignant lesions (foci of cellular alterations (FCA); *n* = 484; Figure 1C) as well as a broad variety of malignant lesions. Malignant neoplastic lesions included the diagnosis of hepatocellular carcinoma (HCC) (*n* = 94), among those 26 early HCCs with prominent vascular pattern (Figure 1D,E); cholangiocarcinoma (CC) (*n* = 70, Figure 1F); undifferentiated tumors (most probably poorly differentiated HCC or CC, *n* = 5; Figure 1G) and neoplastic/malignant tumors of non-hepatic origin (lymphoma, *n* = 10; Figure 1H). Further, 33 lesions presented with a prominent tumorous oval cell proliferation with prominent spindle cells and inflammatory background. These lesions were also classified as neoplastic (Appendix A). Twelve lesions showed histopathological signs indicating neoplasia but had to be termed as unclassifiable due to lack of sufficient histopathological criteria to achieve a final diagnosis. In summary, the vast majority of neoplastic lesions showed a hepatocellular differentiation (*n* = 581), followed by cholangiocellular differentiation (*n* = 73). 

In contrast, 160 (=18.3%) of all lesions had to be classified as non-neoplastic. A wide spectrum of non-neoplastic lesions was identified including hepatocellular hyperplasia (*n* = 11; Figure 2A) and metabolic changes such as macro-/microvesicular steatosis (*n* = 32) or glycogen accumulation (*n* = 25; Figure 2B). Necrosis and/or fibrosis were present in 10 cases (Figure 2C). Interestingly, one fifth of non-neoplastic lesions (*n* = 32) presented with a wide range and spectrum of reactive changes associated with the biliary tree: lesions included bile duct hyperplasia (*n* = 19; Figure 2D), inflammatory bile duct changes, cystic bile duct dilation and cholangiofibrosis (*n* = 13; Figure 2E,F). 

Further, oval cell proliferations, which did not fully classify as true neoplasia, were found in *n* = 23 lesions (Appendix A) as well as inflammation and lobular hepatitis in 31 lesions (Appendix A). A comprehensive overview of neoplastic and non-neoplastic diagnosis of all models is shown in Figure 3.

Models with KRAS mutation presented with a three times higher average number of histological detectable lesions compared to PTEN mutated models (KRAS: *n* = 5.62 (2.17–9.08) vs. PTEN: *n* = 1.66 (0.8–2.50); Figure 4A). Further, KRAS mutation led to a far lower percentage of non-neoplastic lesions (KRAS: 4.1% vs. PTEN: 57%, Figure 4B). However, all models independent of their genetic background developed malignant or premalignant lesions (KRAS: 90.1% of all detectable lesions (81.91–98.31%), PTEN: 39.55% (6.25–98.31%)). With regard to sex or age distribution, no specific correlation could be found.

Both KRAS or PTEN mutated models showed background neoplastic lesions (lymphoma) (KRAS: *n* = 7 vs. PTEN: *n* = 3; Figure 4C) and undifferentiated tumors (KRAS: *n* = 4 vs. PTEN: *n* = 1 Figure 4C) with a slight predominance of the KRAS models in both categories. 

Depending on the genetic background of each model, a wide distribution and incidence of malignant lesions was seen (Figure 4D,E). Cholangiocellular carcinoma (CC) was the most frequent malignant tumor observed in the PTEN/TGFβR2 model (Figure 4C); however, in all other models, foci of cellular alteration (FCA) was the most frequent lesion (Figure 4C,D) followed by hepatocellular carcinoma (HCC).

Analyzing the distribution of non-neoplastic lesions, a huge variety among the different models was detectable (Figure 4F). In the PTEN mutated background, a broad distribution of several lesions throughout the models was present with detection of at least five different non-neoplastic lesions per model. However, when looking at the KRAS mutated background, both models only showed few non-neoplastic lesions in a very low percentage. No hyperplastic changes were observed. 

## 4. Discussion

Genetically engineered mouse models (GEMM) are an important tool in modern biomedical research [1,33,34,35]. However, a precise phenotyping of the newly designed models is often missing which can lead to misinterpretation and misconclusions in the context of other highly relevant findings [11]. Chemically induced rodent models for liver tumors have been widely used for decades; thus these models present with a profound histopathological classification of proliferative and non-proliferative liver lesions [2]. In contrast, not much is known about the distribution of such lesions in GEMM and, unfortunately, attempts to improve this situation are rare. Our data—drawn from a broad variety of genetically engineered mouse models for liver tumorigenesis-identified a wide spectrum of neoplastic and non-neoplastic, as well as benign and malignant, liver lesions comparable to the spectrum reported in chemically induced models. All five evaluated models harbored mutations that are widely used in published literature to study liver tumorigensis in vivo [29,36,37,38,39] reflecting only a small percentage of genetic alterations causing liver tumors [40,41,42,43,44,45,46]. According to our findings, all models were suitable for tumorigenesis research as all reliably developed malignant tumors. KRAS mutated models in direct comparison with all other models used in this study seemed to be more convenient for general research purposes, presenting a robustly high percentage of malignant tumors but a low rate of benign or non-neoplastic lesions. Combining KRAS with PTEN further enhanced the percentage of malignant tumors but not the percentage of non-neoplastic lesions. Thus, overall, GEMM with a KRAS-mutated background might also be appropriate for researchers with little experience in histopathology. However, these models will probably not be able to sufficiently mimic the natural development of liver tumors in diseased organs (e.g., cirrhotic liver, steatosis, fibrosis) which (depending on the biomedical hypothesis these models are used for) warrants critical discussion and careful consideration in data interpretation. In contrast, PTEN mutated models presented with a wide spectrum of neoplastic, both benign and malignant, as well as non-neoplastic, lesions probably more closely mimicking the human situation. Of note, these models may much better resemble the variety of human liver pathology but also consequently request a higher number of mice included in tumor experiments to reliably reproduce results and balance the wide spectrum of histopathological findings. Further, our data clearly show that additional genetic alterations on a PTEN background might lead to extensive changes in the distribution and appearance of histopathological diagnosis. Therefore, these models should only be considered for tumor research purposes if sufficient histopathological knowledge is available in the laboratory or in the collaborative network of the scientist. However, if applied for scientific questions on liver pathologies other than tumorigenesis, these models might be very beneficial. With respect to the non-neoplastic changes, one important finding in our study is that microscopic glycogen accumulation in hepatocytes, a well-known background lesion in rodents with a daytime-dependent effect, can also macroscopically be interpreted as a tumorous lesion. 

We observed a high incidence of foci of cellular alterations (FCA) in all types of GEMM included in this study. In carcinogen-induced models, these lesions have been investigated thoroughly as morphological representation of metabolic aberrations associated with cellular changes [25]. Our findings show that comparable cellular alterations might also occur in genetically modified models of liver carcinogenesis. In carcinogen-induced liver carcinogenesis in rats, different types of FCA have been identified as precursor lesions of malignant hepatocellular neoplasms [24,25]. This has not been proven in genetically modified mouse models. 

Taken together, four main findings of this study highlight the importance of classic H&E-based histology in biomedical research: (i) the number of histologically distinguishable lesions was on average more than twice the number of macroscopically detectable tumorous lesions that were sent for histopathological evaluation, (ii) all lesions—irrespective of their neoplastic or non-neoplastic nature—appeared as a visible “tumor nodule”, (iii) roughly 20% (which means on average every fifth visible tumor nodule) turned out to be of non-neoplastic/reactive origin and (iv) GEMM with different genetic backgrounds may present with an extensive spectrum and distribution of histopathological diagnoses and only slight alterations may change this distribution significantly.

Thus, these results clearly indicate that pure macroscopic descriptions of tumorous lesions or imaging data without corresponding microscopic/histological correlation (or any other more detailed characterization such as molecular characterization) might not be sufficient to fully and correctly characterize (newly designed) mouse models for liver tumorigenesis. 

It is widely discussed that biomedical research struggles with persistent problems in reproducibility [47,48,49]. Our findings, based on simple H&E stains but evaluated by experienced pathologists, fully support this discussion and identify one of the main reasons for reproducibility problems in GEMM research. We believe that our findings are of great significance for all researchers dealing with GEMM in the setting of liver tumorigenesis. We hope that this study might help to create increasing awareness for histopathology-based research approaches as an indispensable, simple and cost-efficient tool regardless of whether they are performed by pathologists (either human or veterinary—both need a specific expertise in this field) or researchers with pathology expertise potentially backed by an experimental pathology core unit [10]. Such an awareness, and continuous thorough characterizations of liver lesions, will further improve the results in GEMM-based liver tumor research. Studies like this should, in general, further encourage scientific journals publishing biomedical results including new GEMM to request (i) correct and comprehensive phenotyping of GEMM and (ii) a histopathological classification of any macroscopically detected lesion/alterations.

## 5. Conclusions

GEMMs in biomedical research have become a widely used elegant tool to study tumorigenesis in vivo. However, appropriate usage of these models and a clear (histological) phenotyping are essential to correct interpretation of results and conclusions.

## Figures and Tables

**Figure 1 cancers-12-02265-f001:**
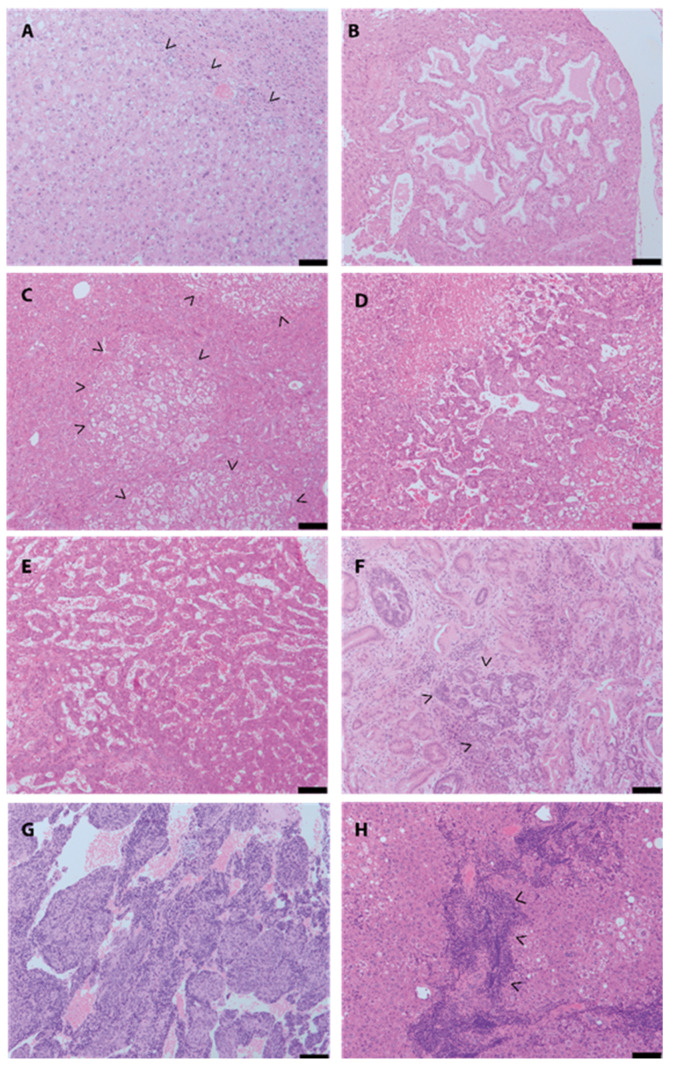
Spectrum of neoplastic lesions identified in genetically engineered mouse models (GEMM) models. (**A**) hepatocellular adenoma; (**B**) bile duct adenoma; (**C**) foci of cellular alterations; (**D,E**) hepatocellular carcinoma, early (**D**) and advanced (**E**) stage, (**F**) cholangiocarcinoma; (**G**) undifferentiated tumors (most probably poorly differentiated hepatocellular carcinoma (HCC)); (**H**) lymphoma. Scale bar: 100 µm, >marks the lesions.

**Figure 2 cancers-12-02265-f002:**
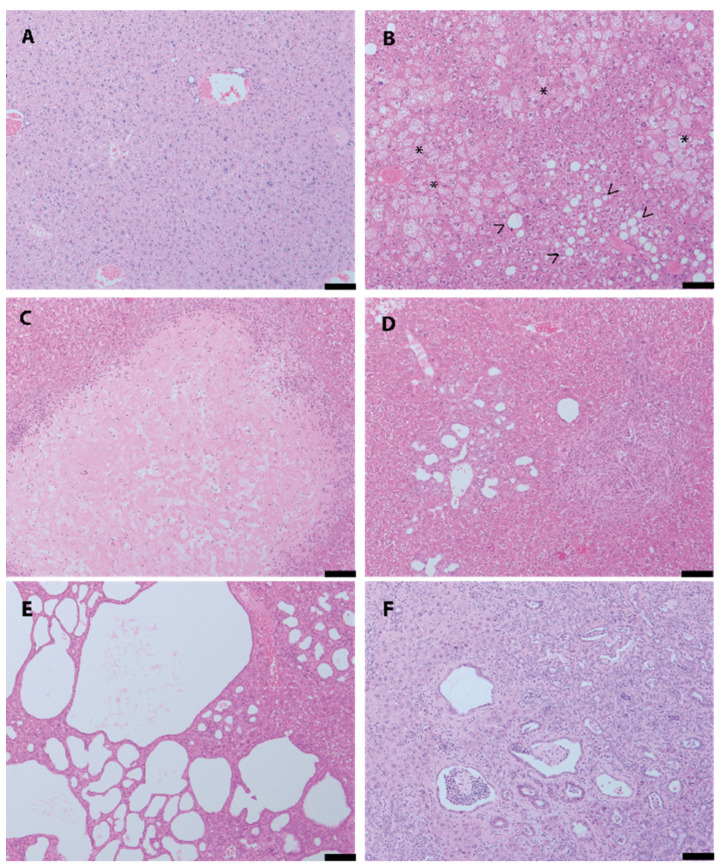
Spectrum of non-neoplastic lesions. (**A**) hepatocellular hyperplasia; (**B**) metabolic changes including macro-/microvesicular steatosis (<) and glycogen accumulation (*); (**C**) fibrosis; (**D**) bile duct hyperplasia; (**E**,**F**) inflammatory bile duct changes including cystic bile duct dilation (**E**) and cholangiofibrosis (**F**). Scale bar: 100 µm.

**Figure 3 cancers-12-02265-f003:**
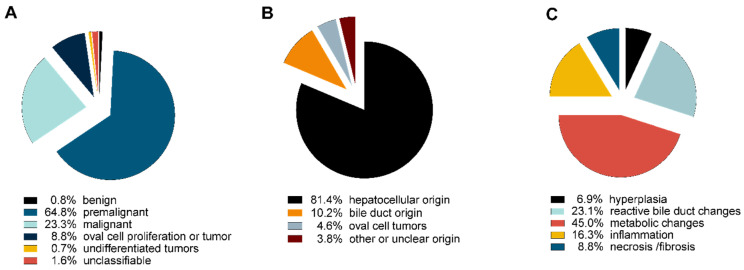
Distribution of observed lesions. (**A**) Distribution of observed lesions; (**B**) histopathological origin of observed neoplastic lesions; (**C**) distribution of non-neoplastic lesions.

**Figure 4 cancers-12-02265-f004:**
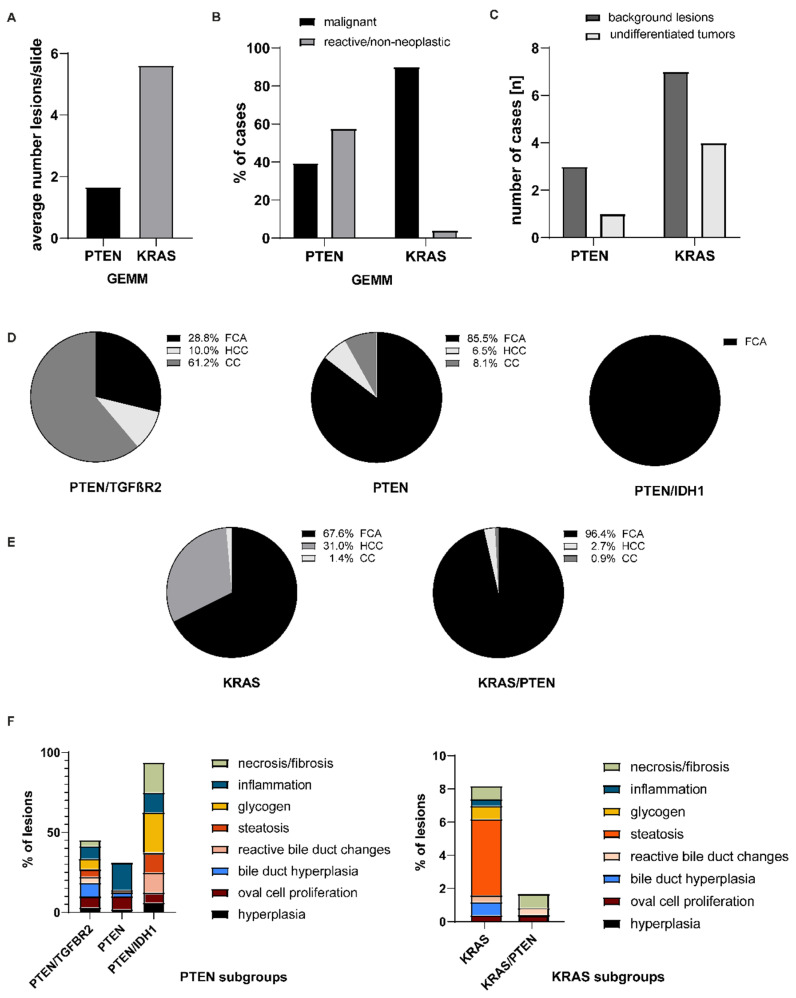
Distribution of diagnosis according to the genetic background. (**A**) Average number of microscopically detectable lesions; (**B**) percentage of non-neoplastic and neoplastic lesions; (**C**) distribution of background lesions and undifferentiated tumors in PTEN and KRAS models; (**D**) distribution of malignant tumors in PTEN mutated subgroups; (**E**) distribution of malignant tumors in KRAS mutated subgroups. (**F**) Distribution of non-neoplastic lesions in PTEN and KRAS subgroups.

**Table 1 cancers-12-02265-t001:** Diagnosis and distribution of lesions in GEMM [%].

	**Lesions**	**FCA**	**HCC**	**CC**	**Undifferentiated**	**Oval Cell Tumor**	**Background Lesion**	**Unclassifiable**	**Bile Duct/Hepatocellular Adenoma**
**GEMM**									
PTEN/TGFβR2		13.8	4.8	29.3	0	0	1.6	3.7	0/0
PTEN		51.7	3.9	4.9	0.5	2.0	0	2.4	0/0.49
PTEN/IDH1		6.3	0	0	0	0	0	0	0/0
KRAS		53.4	24.5	1.1	1.5	7.6	1.9	0	1.1/0.8
KRAS/PTEN		89.4	2.5	0.8	0	5.5	0.9	0	0/0
	**Lesions**	**Hyperplasia**	**Oval Cell Proliferation**	**Bile Duct Hyperplasia**	**Reactive Bile Duct Changes**	**Steatosis**	**Glycogen Accumulation**	**Inflammation**	**Necrosis/Fibrosis**
**GEMM**									
PTEN/TGFβR2		3.2	6.9	8.5	3.7	4.8	6.4	8.0	3.7
PTEN		2.0	8.3	2.4	1.5	17.0	0	0	0
PTEN/IDH1		6.3	6.3	0	12.5	12.5	25.0	12.5	18.8
KRAS		0	0.4	0.8	0.4	4.6	0.8	0.4	0.8
KRAS/PTEN		2.0	8.3	2.4	1.5	17.0	6.8	3.9	1.0

Percentage of lesions according to specific genetic modification.

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
