# Peer review of "Genetically Engineered Mouse Models of Liver Tumorigenesis Reveal a Wide Histological Spectrum of Neoplastic and Non-Neoplastic Liver Lesions"

_cancers, 2020, doi:10.3390/cancers12082265_

Round 1

Reviewer 1 Report

Materials and Methods:

Remove period after 369.

"From" is misspelled line 53.

Please use standardized terminology and form to express transgenic lines (http://www.informatics.jax.org/mgihome/nomen/gene.shtml). 

Please use a standard font and correct capitalization of the university.

Please provide references for "adequate levels of genetic modification/deletion of each mouse LINE" (line 61).

Results:

Spell out "Twelve" (line 84).

Table 1 needs to come near the beginning of this section. This would clarify all the possible results as they are subdivided in subsequent figures. 

Please rephrase "inhomogenous glycogen storage" (lines 96-97 and Fig 2 legend); I have no idea what this means (and I am a pathologist). This is not standard terminology.

"Considering the specific genetic background..." (line 114).

"histologically detectable" (line 115).

Please rephrase "foci of cellular alteration (FCA) was the most frequent tumor" (line 133). FCA are not tumors. Some types (which the authors did not sub-diagnose) are considered premalignant, but it is incorrect to regard them as tumors (swellings).

Figures and Legends:

Figures 1 and 2 should be a full page width and in better focus to allow the changes depicted to be clearly seen by a reader.

Figures 1 and 2 should be annotated to clearly show the different changes described in the narrative for the non-pathologist reader (unless the change is diffuse and takes up the entire frame).

Fig 1D and 1E should be reversed to present the early lesion first.

Fig 2B legend needs "and".

Figure 3 needs to employ different black and white patterns rather than shades of gray which are very difficult to discern and have odd black margins on some edges. Additionally, it would be appropriate to decrease the significant figures behind the decimal point (i.e. "8.8%" rather than "8.84%"), This would simplify the figure and make the important information stand out better.

Fig 3A legend makes no sense. 

Figure 4 needs to employ different black and white patterns in all parts. Gray bars surrounded by a black border are not distinguishable from gray bars (4C).

Table 1 needs to be in black and white with capitalization of categories.

Discussion:

Please rephrase "Especially KRAS mutated models..." (lines 155-157), as this is not a sentence.

Please change "an" to "and" in line 179 and use semicolons between the clauses.

Author Response

We sincerely would like to thank the expert reviewer for her/his very critical and detailed review. This really helped us to improve the manuscript. Following the reviewer´s outline we have addressed the comments by dividing the responses into the specific sections.

Comments on the Materials and Methods section

Comments 1 & 2:

Remove period after 369.

"From" is misspelled line 53.

We have changed that accordingly (marked in “track changes”).

Comment 3:

Please use standardized terminology and form to express transgenic lines (http://www.informatics.jax.org/mgihome/nomen/gene.shtml). 

We have added the specific mutations/transgenic lines (marked in “track changes”).

Comment 4:

Please use a standard font and correct capitalization of the university.

We have changed that accordingly (marked in “track changes”).

Comment 5:

Please provide references for "adequate levels of genetic modification/deletion of each mouse LINE" (line 61).

We have added two more references in the Material and Methods Section.

Comments on the Results section

Comment 6:

Spell out "Twelve" (line 84).

We have changed that accordingly (marked in “track changes”).

Comment 7:

Table 1 needs to come near the beginning of this section. This would clarify all the possible results as they are subdivided in subsequent figures. 

Thank you for this helpful observation. We have changed the manuscript and inserted the table right at the beginning of the results section. As requested by the editorial board we have removed the table from this main text and added it as a separate file (for details please see the section editorial comments #5 in this letter).

Comment 8:

Please rephrase "inhomogenous glycogen storage" (lines 96-97 and Fig 2 legend); I have no idea what this means (and I am a pathologist). This is not standard terminology.

We have rephrased "inhomogenous glycogen storage" to “glycogen accumulation”.

Comment 9 & 10:

"Considering the specific genetic background..." (line 114).

"histologically detectable" (line 115).

We have changed both sentences accordingly.

Comment 11:

Please rephrase "foci of cellular alteration (FCA) was the most frequent tumor" (line 133). FCA are not tumors. Some types (which the authors did not sub-diagnose) are considered premalignant, but it is incorrect to regard them as tumors (swellings).

We have rephrased it into “lesion”. And added a critical discussion on FCA in GEMM in the discussion.

Comments on the Figures and Legends Section:

Comment 12:

Figures 1 and 2 should be a full page width and in better focus to allow the changes depicted to be clearly seen by a reader.

We fully agree with the reviewer! Our original file indeed covers a full page. However, this special format requested by the authors´guidelines for the review process required a change of the format to what the reviewer has now seen.

Comment 13:

Figures 1 and 2 should be annotated to clearly show the different changes described in the narrative for the non-pathologist reader (unless the change is diffuse and takes up the entire frame).

Excellent idea. We added arrowheads (>) and asterix (*) into figures 1 and 2 to better mark and highlight the different lesions.

Comment 14 & 15:

Fig 1D and 1E should be reversed to present the early lesion first.

Fig 2B legend needs "and".

We have changed the figures 1D and 1E and included “and” in figure 2B (figure legends). For clearer presentation we have removed the old version of the figure from the revised manuscript (text is presented in “track changes”).

Comment 16:

Figure 3 needs to employ different black and white patterns rather than shades of gray which are very difficult to discern and have odd black margins on some edges. Additionally, it would be appropriate to decrease the significant figures behind the decimal point (i.e. "8.8%" rather than "8.84%"), This would simplify the figure and make the important information stand out better. Fig 3A legend makes no sense. 

We agree with the reviewer that reducing the information to one figure behind the decimal point is a good idea and helps to highlight the findings. Concerning the comment on the colors we understand that different shades of grey might be difficult to distinguish. We have therefore changed the panels with smaller areas into color.

Comment 17:

Figure 4 needs to employ different black and white patterns in all parts. Gray bars surrounded by a black border are not distinguishable from gray bars (4C).

We have changed some of the grey patterns in Figure 4A-E to create a better contrast between the different groups. For Figure 4F we agree with the reviewer that the different subgroups are difficult to distinguish. That´s why we have changed this to color.

Comment 18:

Table 1 needs to be in black and white with capitalization of categories.

We have changed that accordingly to the reviewer´s suggestion and removed the table from the main text according to the editorial suggestion. The table will be sent as a separate file.

Comments on Discussion Section:

Comment 19 & 20:

Please rephrase "Especially KRAS mutated models..." (lines 155-157), as this is not a sentence.

Please change "an" to "and" in line 179 and use semicolons between the clauses.

We have changed both sentences accordingly.

Reviewer 2 Report

This manuscript addresses an important point that is often overlooked in the 'liver cancer' field involving the histopathological analysis of tumours. Most studies report macroscopic tumours as cancer, which is incorrect. This study demonstrates the heterogeneity found within GEMMs and between different GEMMs. The only thing that is unfortunate is that KRAS and PTEN are low frequency mutations in HCC, so these models are not really very good for garden variety HCC studies. If the authors could add additional animal models with more relevance to mutations present in HCC, then it would be much more interesting. I do not agree with the authors suggestion that KRAS makes a good model for new investigators to study HCC. The authors did not mention whether the models were age-matched or controlled in some other way. There is evidence that HCA turns into HCC so the age of animals could make a difference in incidence of each. Were the mice fed the same diet? What gender? More details of the animals, sex, diet is required. 

Author Response

General comments

We really thank the editors and reviewers for this very positive feedback including a critical and detailed review section overall highlighting the relevance of our work. As suggested by the editors and the reviewers we have specially addressed the comments of each sections by a point-to-point response to reviewers letter.

Comment 1:

This manuscript addresses an important point that is often overlooked in the 'liver cancer' field involving the histopathological analysis of tumours. Most studies report macroscopic tumours as cancer, which is incorrect. This study demonstrates the heterogeneity found within GEMMs and between different GEMMs.

First of all, we would sincerely like to thank the reviewer for his/her overall very positive but critical feedback. We fully agree with the reviewer that our work addresses a totally overlooked area in the field of biomedical research and we hope that our work can contribute to improve this situation.

Comment 2:

The only thing that is unfortunate is that KRAS and PTEN are low frequency mutations in HCC, so these models are not really very good for garden variety HCC studies. If the authors could add additional animal models with more relevance to mutations present in HCC, then it would be much more interesting.

In this point we also fully agree with the reviewer that the mentioned mutations are not the most common ones observed in HCC. However, our work does not focus on the process of hepatocarcinogenesis solely but on models used for liver tumorigenesis in general. Further, the models used in our study to some extent reflect scientific reality as all models are used to address questions on liver tumorigenensis. To address the reviewer´s comment best we have added a new part in the discussion section of the revised manuscript.

Comment 3:

I do not agree with the authors suggestion that KRAS makes a good model for new investigators to study HCC.

We apologize for the misconclusion. Our aim was to compare the models used in our study in terms of different aspects that might be relevant for researchers (e.g. feasibility). We agree that there are other models to better investigate hepatocarcinogenesis. To make this point clearer we have rewritten the discussion section accordingly.

Comment 4:

The authors did not mention whether the models were age-matched or controlled in some other way. There is evidence that HCA turns into HCC so the age of animals could make a difference in incidence of each. Were the mice fed the same diet? What gender? More details of the animals, sex, diet is required. 

To address this comment we have included a new table with additional information on sex distribution and age. In direct comparison we do not see a significant correlation between one of these parameters and the distribution of observed lesions. We included this in the result sections. Furthermore, all mice were kept under same laboratory conditions with standard twelve hour day/night cycle and food (standard diet) and water ad libitum. We added this information in the Material and Methods Section.

Round 2

Reviewer 2 Report

The authors have made changes to improve the manuscript. I have no further comments.